# A comprehensive guide to CAN IDS data and introduction of the ROAD dataset

Miki E. Verma[1]☯, Robert A. Bridges[2]☯, Michael D. Iannacone[2], Samuel C. Hollifield[2], Pablo Moriano[3]*, Steven C. Hespeler[3], Bill Kay[3,4], Frank L. Combs[5]

**1** Rewiring America, United States of America, **2** Cyber Resilience and Intelligence Division, Oak Ridge National Laboratory, Oak Ridge, TN, United States of America, **3** Computer Science and Mathematics Division, Oak Ridge National Laboratory, Oak Ridge, TN, United States of America, **4** Computational Mathematics, Pacific Northwest National Laboratory, Richland, WA, United States of America, **5** Electrification and Energy Infrastructures Division, Oak Ridge National Laboratory, Oak Ridge, TN, United States of America

☯ These authors contributed equally to this work.
* moriano@ornl.gov

**Data Availability Statement:** Dataset available at: https://zenodo.org/records/10462796 and https://0xsam.com/road/.

## Abstract

Although ubiquitous in modern vehicles, Controller Area Networks (CANs) lack basic security properties and are easily exploitable. A rapidly growing field of CAN security research has emerged that seeks to detect intrusions or anomalies on CANs. Producing vehicular CAN data with a variety of intrusions is a difficult task for most researchers as it requires expensive assets and deep expertise. To illuminate this task, we introduce the first comprehensive guide to the existing open CAN intrusion detection system (IDS) datasets. We categorize attacks on CANs including fabrication (adding frames, e.g., flooding or targeting and ID), suspension (removing an ID's frames), and masquerade attacks (spoofed frames sent in lieu of suspended ones). We provide a quality analysis of each dataset; an enumeration of each datasets' attacks, benefits, and drawbacks; categorization as real vs. simulated CAN data and real vs. simulated attacks; whether the data is raw CAN data or signal-translated; number of vehicles/CANs; quantity in terms of time; and finally a suggested use case of each dataset. State-of-the-art public CAN IDS datasets are limited to real fabrication (simple message injection) attacks and simulated attacks often in synthetic data, lacking fidelity. In general, the physical effects of attacks on the vehicle are not verified in the available datasets. Only one dataset provides signal-translated data but is missing a corresponding "raw" binary version. This issue pigeon-holes CAN IDS research into testing on limited and often inappropriate data (usually with attacks that are too easily detectable to truly test the method). The scarcity of appropriate data has stymied comparability and reproducibility of results for researchers. As our primary contribution, we present the Real ORNL Automotive Dynamometer (ROAD) CAN IDS dataset, consisting of over 3.5 hours of one vehicle's CAN data. ROAD contains ambient data recorded during a diverse set of activities, and attacks of increasing stealth with multiple variants and instances of real (i.e. non-simulated) fuzzing, fabrication, unique advanced attacks, and simulated masquerade attacks. To facilitate a benchmark for CAN IDS methods that require signal-translated inputs, we also provide the signal time series format for many of

**Funding:** This manuscript has been authored by UT-Battelle, LLC under ContractNo. DE-AC05-00OR22725 with the U.S. Department of Energy. The publisher, by accepting the article for publication, acknowledges that the U.S. Government retains a non-exclusive, paid up, irrevocable, world-wide license to publish or reproduce the published form of the manuscript, or allow others to do so, for U.S. Government purposes. The DOE will provide public access to these results in accordance with the DOE Public Access Plan (http://energy.gov/downloads/doe-public-access-plan). This research was sponsored in part by Oak Ridge National Laboratory's (ORNL's)Laboratory Directed Research and Development program and by the DOE. There was no additional external funding received for this study. The funders had no role in study design, data collection and analysis, decision to publish, or preparation of the manuscript.

**Competing interests:** The authors have declared that no competing interests exist.

the CAN captures. Our contributions aim to facilitate appropriate benchmarking and needed comparability in the CAN IDS research field.

# 1 Introduction

Modern vehicles are increasingly drive-by-wire, relying on continual communication of small computers called electronic control units (ECUs). Nearly ubiquitous in modern vehicles, Controller Area Networks (CANs) facilitate the data exchange among ECUs by providing a common network with a standard protocol. While lightweight and reliable, the CAN standard has well-known security flaws, lacking authentication, encryption, and other important security features. Furthermore, attack vectors to intra-vehicle CANs are growing in scope as vehicles are increasingly offering channels of connectivity. While exploitation of the CAN bus in previous works is often implemented directly, e.g., by mandatory on-board diagnostics II (OBD-II) ports [7, 8], successful attacks to vehicle CANs also can occur indirectly/remotely through a variety of vehicle interfaces, such as wireless communication channels [9, 10].

Consequently, CAN security and vulnerability research has accelerated, with most literature focused on proving how "hackable" vehicles are [7–14], or proposing novel CAN intrusion detection systems (IDSs) [5, 15–17]. CAN IDS research has grown rapidly, suffering from an inability to reproduce or replicate and compare methods. As a result, proposed detection techniques are often not tested on appropriate data due to lack of availability. E.g., Hossain et al. [18] simulate attacks in real CAN data by adding frames in order to validate an LSTM-based CAN IDS. In theory their IDS may detect much more subtle attacks, e.g., masquerade attacks (see Sec. 2.2), but without available data, this cannot and was not tested. Further, using simulated data or attacks limits fidelity compared to real vehicular CAN data with real, and physically verified, attacks.

To address this problem, we introduce the Real ORNL Automotive Dynamometer (ROAD) dataset, a novel CAN IDS dataset comprised of real automotive CAN data. ROAD contains CAN data from the vehicle during ambient driving including a wide variety of driver activities. The dataset has labeled attacks ranging from easy to difficult to detect. More specifically, ROAD includes multiple variations of real (i.e. non-simulated) fuzzing, fabrication, unique advanced attacks, and simulated masquerade attacks. The goal is to allow appropriate testing and comparable benchmarking of CAN IDS.

To this end, we also provide a thorough guide to all publicly available CAN IDS datasets to aid researchers in selecting and the most appropriate dataset for testing their method. Our survey of previous datasets surveys the publicly-available datasets suitable for CAN IDS testing. We provide for each dataset their references details to help researchers, in particular, data characteristics (real/synthetic, raw CAN and/or signal-translated data, number of vehicles, total time) and attack characteristics (what types of fabrication, suspension, masquerade, or other are present, real/simulated, and whether the attacks are identifiable with simple timing-based methods). See Tables 1–3. We also include a discussion of the uses of the datasets, and findings from our in-depth data analytics on each dataset.

The remainder of this paper is organized into the following sections. The introduction provides the reader with an overview of the state of CAN IDS research. Here, we illustrate two major roadblocks prohibiting this research from advancing and focuses mainly on highlighting the dearth of quality data. We point out the consequences that scarcity of data is having on the community and map them directly to this paper's contributions. Section 2 provides necessary background on CAN protocol and vehicle attack terminology. In particular, we partition attacks into categories fabrication attacks (which add frames to the bus, e.g., DOS and fuzzing),

**Table 1. Open CAN IDS datasets.**

| | Dataset | Organization | Year | Dataset URL |
|---|---|---|---|---|
| 1 | CAN Intrusion (OTIDS) [1] | HCRL | 2017 | http://ocslab.hksecurity.net/Dataset/CAN-intrusion-dataset |
| 2 | Survival Analysis for Auto. IDS [2] | HCRL | 2018 | http://ocslab.hksecurity.net/Datasets/survival-ids |
| 3 | Car Hacking for Intrusion Detection [3, 4] | HCRL | 2018 | http://ocslab.hksecurity.net/Datasets/CAN-intrusion-dataset |
| 4 | SynCAN [5] | Bosch | 2019 | https://github.com/etas/SynCAN |
| 5 | Auto. CAN Bus Intrusion v2 [6] | TU Eindhoven | 2019 | https://doi.org/10.4121/uuid:b74b4928-c377-4585-9432-2004dfa20a5d |
| 6 | Can Log Infector | CrySyS Lab | 2020 | https://www.crysys.hu/research/vehicle-security/ |
| 7 | **ROAD CAN Intrusion Dataset** | ORNL | 2020 | https://zenodo.org/records/10462796 |

All datasets include ambient data for the given vehicle(s) in addition to attack data.

**Table 2. Open CAN IDS datasets' metadata.**

| | Dataset | Real / Synthetic | Raw CAN | Signal Translated | Type | Total Time |
|---|---|---|---|---|---|---|
| 1 | CAN Intrusion (OTIDS) [1] | Real | ✓ | | 1 vehicle | 1h 8m 0s |
| 2 | Survival Analysis for Auto. IDS [2] | Real | ✓ | | 3 vehicles | 0h 12m 24s |
| 3 | Car Hacking for Intrusion Detection [3, 4] | Real | ✓ | | 1 vehicle | 7h 38m 28s |
| 4 | SynCAN [5] | Synthetic | | ✓ | – | 24h 45m 7s |
| 5 | Auto. CAN Bus Intrusion v2 [6] | Real & Synthetic | ✓ | | 2 vehicles, 1 testbed | 1h 42m 51s |
| 6 | Can Log Infector | Real | ✓ | | CAN paired with GPS | 0h 30m 46s |
| 7 | **ROAD CAN Intrusion Dataset** | Real | ✓ | ✓ | 1 vehicle | 3h 27m 10s |

Total time of ROAD CAN data is 3h 27m 10s. This does not include the signal translations of the same data. ROAD includes 13 masquerade attack files, which are identical to the corresponding targeted ID captures but with the ambient frames of the target ID removed during the attack period to simulate the masquerade. ROAD total CAN data time without the 13 masquerade attacks is 3h 16m 13s.

**Table 3. Open CAN IDS datasets' attack metadata.**

| No. | Dataset | Fabrication | Suspension | Masquerade | Other | Attack Types[1] T.T. | Attack Types[1] T.O. |
|---|---|---|---|---|---|---|---|
| 1 | CAN Intrusion (OTIDS) [1] | Real; DoS, Fuzzing | – | – | Remote frame impersonation | 2 | 1[2] |
| 2 | Survival Analysis for Auto. IDS [2] | Real (flooding); DoS, Fuzzing, Targeted ID | – | – | – | 3 | 0 |
| 3 | Car Hacking for Intrusion Detect. [3, 4] | Real (flooding); DoS, Fuzzing, Targeted ID[3] | – | – | – | 4 | 0 |
| 4 | SynCAN [5] | – | Simulated | Simulated | – | 2 | 3 |
| 5 | Auto. CAN Bus Intrusion v2 [6] | – | Simulated | Simulated | – | 6 | 1 |
| 6 | Can Log Infector | – | – | Provides code to simulate 7 types [4] | – | 0 | 7 |
| 7 | **ROAD CAN Intrusion Dataset** | Real (flooding) Fuzzing, (flam) Targeted ID | – | Simulated | Unique advanced attacks | 5 | 5 |

[1]. "(resp. "T.O.") refer to Timing Transparent (Timing Opaque) meaning the attack does (does not) alter normal timing characteristics.

[2]. The sole advanced attack, the "Impersonation Attack," may not actually be useful for researchers developing an IDS (see description in Sec. 3).

[3]. Due to the crude method of creating simulated injections (see Sec. 3), involving modifying timestamps, classification is less clear.

[4]. Does not include attack data, but has several ambient captures and a Python script for modifying logs to create simulated attacks.

suspension attacks (which prevent frames from being sent thereby removing them from the bus), masquerade attacks (replacing legitimate frames with malicious frames), and other. Specific definitions of types of these attacks are discussed. Section 3 comprises the survey, analysis and discussion of all previous CAN attack datasets. Section 4 introduces our new CAN attack dataset, and Section 5 concludes this work.

## 1.1 The growth and state of CAN IDS research

CAN IDS works can be grouped by CAN features distinctive to the data, yielding the following five categories:

**Frequency/Timing-Based**: Regards the timing or sequencing of arbitration IDs [17, 19–23]

**Payload-Based**: Considers the data frame (message contents) as a string of bits, without explicitly recovering the signals these bits represent [16, 24–30]

**Signal-Based**: Requires first decoding raw data field bits into constituent signals, and uses time series' of signal values as inputs [5, 17, 31–37]

**Physical Side-Channel**: Uses physical layer attributes (e.g., voltage) [15, 38–40]

**Other**: Includes works that do not fall into the above categories (e.g., using rules to guarantee specific characteristics of the CAN messages are followed [41, 42]).

CAN IDS methods can also be dichotomized into Specification-Based and Machine-Learning-Based (ML-based) methods. The former relies on manually crafted rules to detect if the CAN protocol or the rules of the CAN encodings for a particular vehicle are broken [41, 43, 44]. For example, a rule that checks whether a particular ID's data field has the correct number of bytes. ML-based approaches instead employ an algorithm that is trained on observed data to "learn" mapping features of the CAN data that are either considered anomalous or malicious [1, 5, 45]. Unsupervised learning examples include, learning the expected timings of each message to identify injected messages [19] and using side-channel techniques to fingerprint a node on the CAN bus with the goal of identifying a rogue node in the future [15]. Supervised learning examples include training a classifier on known malicious and ambient messages [24, 46]. The target of this work is to survey current CAN IDS datasets and present a new dataset to assist ML-based detectors that reside in the first three categories (Frequency/Timing-Based, Payload-Based, Signal-Based).

Four recent in-vehicle IDS surveys itemize many CAN IDS works, and these surveys show the growth and progress of this field [47–51]. Note that these surveys (even when combined) are not comprehensive but provide a representative sample of the existing CAN security methods. The two recent surveys by Wu et al. [47] and Lokman et al. [48] demonstrate both the increasing pace and asymmetric growth of the field in terms of the five categories described above. In [51], Rajapaksha, et al. review and categorize recent AI-based IDSs for securing In-Vehicle Networks (IVNs), with a particular focus on the CAN bus, offering valuable insights into effective AI algorithms and their applications in the context of IVN cybersecurity. The study highlights the superior detection capabilities of deep learning algorithms, particularly LSTM and GRU-based ensemble models, in the context of IVN security, along with promising trends in unsupervised learning, transfer learning, and the potential to combine physical characteristics with CAN data frame features for enhanced attack detection. A quick meta-analysis demonstrates the recent trends (up to 2023) of CAN IDS papers and shows the current interest in the topic. Fig 1 highlights these trends based on yearly publications and category breakdown, showcasing a steady yearly trend of CAN IDS publications since 2014.

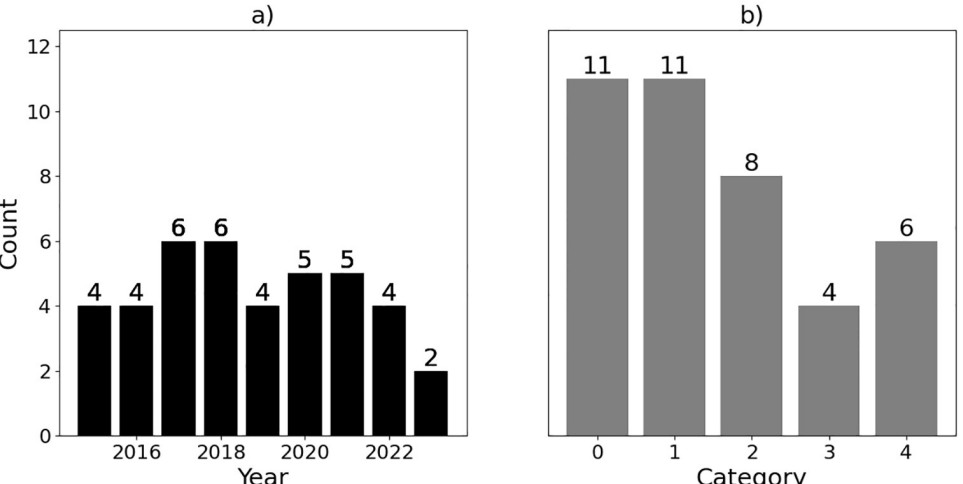

**Fig 1. Papers published in peer reviewed journals based on: a) yearly trend of CAN IDS research and b) frequency of of CAN IDS category; 1) frequency/timing-based, 2) payload-based, 3) signal-based, 4) physical side-channel, and 5) other.**

While the number of publications in the CAN security domain, especially in IDS research, has grown appreciably in the past few years, IDS research is significantly hindered by two major issues: (1) proprietary CAN signals (not the focus of this work that has been approached by CAN reverse engineering frameworks) and (2) lack of high-quality, publicly available, real CAN data with advanced attacks present (the focus of this work). We detail work on CAN reverse engineering in Section 1.1.1.

**1.1.1 CAN reverse engineering problem.** The asymmetric growth in the field—in particular the disproportionate number of publications on methods that are timing/frequency-based (and to a lesser extent payload-based) as compared to signal-based—is a direct result of the proprietary CAN signal encodings issue. Original equipment manufacturers (OEMs, e.g., Subaru, Ford) of passenger vehicles hold secret their proprietary encodings of signals in the CAN data fields and vary the encodings across models. Consequently, though researchers can easily add a node to monitor and send CAN messages on most vehicles, the data is not understandable. Thus, most have focused on methods that do not require knowledge of signal encodings. Indeed a few researchers have paired with OEMs or done some manual reverse engineering to obtain and develop IDSs based on the decoded CAN signals, though these developments are not vehicle-agnostic.

Although the proprietary CAN signal problem is not addressed in this paper, it is necessary context for the issue of availability of real automotive CAN data with advanced attacks present. Partially reverse engineered signal mappings are emerging online, and a small subfield has emerged with the goal of automating the reverse engineering of signals from automotive CAN data [53, 55–58, 61]). Verma et al. [61] provide a comprehensive treatment of the problem as well as a survey of the previous work. In particular, the signal-reverse engineering problem can be broken into four sub-problems: (1) tokenization—identifying the signal boundaries within the data field (e.g., in a 64-bit data field, bits 1-8 may encode wheel speed, bit 9 a binary indicator of cruise control, bit 10 unused, bits 12-16 a temperature signal, . . .); (2) endianness—for signals crossing a byte boundary the order of the bytes is needed; (3) integer-to-binary encoding—usually base 2 or 2's complement encoding is used; (4) interpretation—an affine mapping of each signal to achieve the correct units and labeling the signal with what it communicates

**Table 4. Automotive CAN signal reverse engineering works.**

| | Tokenization | Endianness | Encoding | Interpretation |
|---|---|---|---|---|
| Jaynes et al. (2016) [52] | ○ | ○ | ○ | ◐ |
| Markowitz & Wool (2017) [53] | ● | ○ | ○ | ○ |
| Huybrechts et al. (2018) [54] | ◐ | ○ | ○ | ◐ |
| Nolan et al.'s TANG (2018) [55] | ● | ○ | ○ | ○ |
| Marchetti & Stabili's READ (2018) [56] | ● | ○ | ○ | ○ |
| Verma et al.'s ACTT (2018) [57] | ● | ○ | ○ | ● |
| Pesé et al. LibreCAN (2019) [58] | ● | ○ | ○ | ● |
| Young et al. (2020) [59] | ○ | ○ | ○ | ◐ |
| Buscemi et al. (2021) CANMatch [60] | ● | ● | ○ | ● |
| Verma et al. (2021) CAN-D [61] | ● | ● | ● | |

(e.g., speed in mph). Table 4 itemizes previous CAN reverse engineering works and the portions of the four sub-problems to which they contribute. For a comprehensive survey on CAN reverse engineering, we refer the reader to prior work by Buscemi et al. [62] for further details. Overall, signal reverse engineering can facilitate CAN IDS research and, more generally, may enable a wide variety of downstream automotive technologies, e.g., OpenPilot aftermaket driver assistance technology https://comma.ai/.

## 1.2 Problem addressed

By *real* automotive CAN data we mean a CAN capture from a vehicle. By *synthetic* automotive CAN data we mean data generated by a process designed to emulate a real automobile's CAN. Further, we use the term *real attack* to refer to actual tampering with messages on a CAN, e.g., by adding a node that sends frames, augmenting a node to send malicious frames, or by removing a node that would, in normal conditions, send frames. A *simulated attack* refers to a method to augment a CAN log post collection.

Such data is unavailable for four reasons. First, it is difficult and time-consuming to produce, with the exception of a few basic attacks. Due to available software both open-source, e.g., SocketCAN (https://python-can.readthedocs.io/en/master/interfaces/socketcan.html), CANutils https://github.com/linux-can/can-utils and proprietary, e.g., CANalzyer (https://www.vector.com/int/en/products/products-a-z/software/canalyzer) and VehicleSpy (https://intrepidcs.com/products/software/vehicle-spy), and OBD-II access to many vehicle CANs, reading and sending arbitrary messages on the bus is easy. However, removing/overwriting legitimate messages or sending meaningful messages with a targeted effect is very difficult and is often a per-vehicle endeavor as the former requires advanced hacking skills and the latter involves reverse engineering signals (issue (1)). Thus, CAN data with ambient traffic or crude fabrication attacks are readily available, but real CAN data with subtle attacks or attacks with a targeted physical effect is scarce.

Second, producing realistic CAN attack data carries inherent risks to the passengers, bystanders, and to the vehicle itself. Dynamometers or dedicated tracks, which allow driving in a safe and controlled laboratory environment, can be used, but such facilities are large investments are unavailable to most researchers. Furthermore, risks of permanent damage loom (e.g., "bricking" an ECU).

Third, disclosure of sensitive information is an inhibitor. OEMs consider their CAN encodings intellectual property. Additionally, responsible vulnerability disclosure may be necessary if new attacks are discovered, which at a minimum delays release of data. Further, releasing

data with targeted attacks may be viewed unfavorably by OEMs, resulting in lawsuits if not handled responsibly. In short, developing subtle attacks can be prohibitively expensive as researchers must have a dedicated modern vehicle for study, appropriate facilities for safety, access to deep offensive security expertise, and potentially legal support.

To our knowledge, there are currently six publicly available vehicle CAN datasets with labeled attacks (see Table 1). Likely due to the inherent difficulties in producing real CAN attack data described above, in these datasets, the only real attacks in present real data are fabrication attacks (by message injection)—all other attack captures are either real data with simulated attacks, or are entirely composed of synthetic data. All have significant limitations when supporting CAN IDS development. Fabrication attacks are generally simple to detect with timing-based methods and are thus limited in scope. Due to the complex dynamics of the broadcast CAN protocol, the simulated CAN attacks ignore aberrations in message timing, content, and presence that naturally occur, and therefore change data quality in unknown ways. Further, physical verification of the effect of the simulated attack on the vehicle is not possible. Succinctly, there is no publicly available, real CAN data with labeled attacks that is of sufficient quality to permit assessment of many CAN IDS methods.

## 1.3 Consequences

A survey by Loukas et al. [49] classifies 17 surveyed automotive CAN IDS papers by utilizing the following evaluation methods: "analytical" (theoretical only, no evaluation on data), "simulation" (evaluated on synthetic CAN data or real CAN data with simulated attacks), and "experimental" (evaluated on real CAN data with real attacks). The distribution of the surveyed papers evaluated throughout the article is: analytical *(3)*, simulated *(8)*, experimental *(6)*. We believe this percentage and number of IDS evaluation with real CAN data is too small.

A second consequence is that CAN IDS works are not comparable, or at least not compared. Rajbahadur et al. [50] surveys an even larger set of papers, e.g. with a much wider scope of "Anomaly Detection for Connected Vehicle Cybersecurity"). The investigators found that:

> Much of the research is performed on simulated data (37 out of the 65 surveyed papers) . . . much of the research does not evaluate the newly proposed techniques against a baseline (only 4 out of the 65 surveyed papers do so), which may lead to results that are difficult to quantify.

This also reinforces the findings of Loukas et al. regarding synthetic data/simulated attacks. It appears CAN IDS contributions come from researchers with a wide variety of backgrounds. While this milieu provides a diverse set of approaches, the area suffers by lacking a uniform body of knowledge, and the lack of depth seems to inhibit the steady development of ideas and systematic, quantifiable progress. To again quote Rajbahadur et al.,

> The varied use and scattered publication of anomaly detection [for connected vehicle cybersecurity] research has given rise to a sprawling literature with many gaps and concerns . . . we urge researchers to address these identified shortcomings.

To summarize, quantifiable comparison across competing and complementary IDS methods is currently not possible. Standardized datasets are necessary for head-to-head comparisons and for replicability, or better, reproducibility. To continue to progress with rigor, the CAN IDS research community needs to produce and adopt a publicly shareable collection of CAN datasets with labeled attacks. This sentiment was reiterated and acted on by Hanselmann et al. [5] in their recent CAN IDS work:

To the best our knowledge, there is no standard data set for comparing methods. We try to close this gap by evaluating our model on both real and synthetic data, and we make the synthetic data publicly available. We hope that this simplifies the work of future researchers to compare their work with a baseline.

Finally, we find that IDSs are often evaluated against inappropriate test data. For example, IDSs promising detection of advanced, subtle attacks are tested only on CAN data with exceptionally noisy attacks, or works use attacks that disrupt timing, then ignore timing in evaluation to test payload-based detection. In order to not disparage other IDS works, we cite our own insufficient evaluation of our proposed CAN IDSs as examples [63, 64]. The consequence is that many promising IDS methods, which are excessive for the easily detectable attacks in currently available data, are never truly evaluated on the more advanced attacks they target.

We add to these cries for a more systematic and rigorous progression of CAN IDS research.

## 1.4 Contributions

To address the problem at hand, we provide a comprehensive guide to the publicly available CAN datasets that contain labeled attacks. Our treatment includes datasets with simulated attacks and code bases for simulating attacks in post-processing of ambient data. We itemize these datasets, their download links, citations, metadata (real or synthetic, types, duration), and attack metadata (fabrication, suspension, masquerade, other) clearly in Tables 1–3 for ease of comparison and reference. As most of the public datasets are not accompanied by detailed descriptions nor publications, we performed quality analysis investigations on both the data and documentation of each previously released CAN dataset. In this work, we provide a detailed description of the data, a discussion to illuminate the benefits and drawbacks of each dataset, and recommendations for appropriate use of each dataset when developing a CAN IDS.

Next, we leverage the vehicle and dynamometer resources of Oak Ridge National Laboratory (ORNL) to produce and document the ROAD dataset, collected from a passenger vehicle with a variety of real and simulated CAN attacks. The dataset provides ample (3 hours) training data with no attacks collected when driving on actual roads (not dynamometer). This dataset provides the following real attacks: a fuzzing fabrication attack, many targeted fabrication attacks that are maximally stealthy (manipulating only the necessary portions of the data field and sending a single manipulated message per ambient message of the same ID), and two advanced attacks that include no fabricated (injected) messages. Each attack was physically verified, that is, we observed the effect it has on the vehicle. For each targeted injection attack, we also include an augmented CAN capture by deleting the targeted ambient message to simulate a masquerade attack.

The ROAD dataset fills gaps in the available CAN IDS datasets. The fabrication attacks isolate a targeted ID and send a single frame just after the ambient frame (see Section 4.1.2 This is the most stealthy fabrication attack possible and is not present in of the fabrication attacks of other datasets. This allows the most thorough testing of detectors of fabrication attacks (e.g. timing-based or AID-sequence-based detectors). Secondly, by omitting only the targeted IDs from these attacks we simulate a masquerade attack, which is not present in other datasets. Thirdly, we include advanced attacks which is unique only to ROAD. These latter two attack types allow ROAD to uniquely test payload-based detectors seeking more sophisticated attacks than fabrication attacks. By design the ROAD dataset's attacks increase in difficulty to detect, which is not present in previous datasets. Overall, ROAD allows appropriate testing of more

**Table 5. Logs in ROAD CAN intrusion detection dataset.**

| Log name | Modified | # Logs |
|---|:---:|:---:|
| Accelerator Attack (In Drive) | | 2 |
| Accelerator Attack (In Reverse) | | 2 |
| Correlated Signal Fabr. Attack | | 3 |
| Correlated Signal Masq. Attack | ✓ | 3 |
| Fuzzing Attack | | 3 |
| Max Engine Coolant Temp Fabr. Attack | | 1 |
| Max Engine Coolant Temp Masq. Attack | ✓ | 1 |
| Max Speedometer Fabr. Attack | | 3 |
| Max Speedometer Masq. Attack | ✓ | 3 |
| Reverse Light Off Fabr. Attack | | 3 |
| Reverse Light Off Masq. Attack | ✓ | 3 |
| Reverse Light On Fabr. Attack | | 3 |
| Reverse Light On Masq. Attack | ✓ | 3 |
| Dynamometer Various Ambient | | 10 |
| Road Various Ambient | | 2 |

advanced detectors and facilitating head-to-head comparisons of the wide variety of proposed CAN IDS methods. In all, 33 attack CAN captures are provided. See Table 5.

By using the CAN-D method [61] to convert the raw CAN data to signals, we provide signal-translated time series alongside many of the CAN captures. Our aim is to provide an open, realistic, and verified CAN dataset for benchmarking CAN IDSs that take either raw CAN data or decoded signals' time series as input. This is perhaps the highest fidelity CAN IDS dataset currently available in that all data and attacks were captured from a real vehicle and all attacks are physically verified. It is also the most comprehensive in terms of quantity and diversity of attacks included. Notably, no other real CAN data has fabrication attacks with the stealth of ROAD (using only a single injected frame between ambient frames of the targeted ID and manipulating only a portion of the data field necessary). No other dataset provides both original and translated signals that correspond to raw CAN data (both ambient and attacks).

## 2 CAN basics and attack terminology

### 2.1 CAN protocol

CAN is a message-based protocol standard [65] that defines the first two Open Systems Interconnection (OSI) layers (physical and data link). Using this protocol, ECUs (e.g., Power Control Module [PCM], Antilock Braking System [ABS]) continually broadcast data frames with information relating to the current state of the vehicle. A standard CAN data frame (or packet), depicted in Fig 2, contains several fields, of which two are relevant for the scope of this paper: the 11-bit Arbitration ID, and the 64-bit Data field.

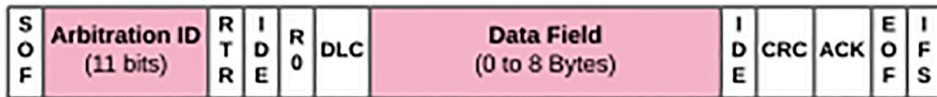

**Fig 2. CAN data frame: The two primary components are the Arbitration ID used for message identification and arbitration (prioritizing messages) and the data field, containing up to 8 bytes of message contents.**

The *Arbitration ID*, or simply ID, is the message header that identifies the frame and is used for arbitration, the process by which frames are prioritized when multiple ECUs concurrently transmit—the lower the ID, the higher the priority. The RTR bit is an indicator of a remote frame. Any ECU can request the data on an ID by sending the ID and the RTR bit indicating the request. This remote frame would be immediately followed by a response with the requested ID and data. The *Data Field* contains the actual message contents of up to 8 bytes, where each distinct piece of information carried in the message is called a signal. CAN frames with the same ID encode the same set of signals in the same format and are usually sent with a fixed frequency to relay updated signal values. In general, each ECU is assigned a set of IDs that only it transmits. For example, the PCM may transmit: ID `0x102` with data field containing engine RPM, vehicle speed, and odometer signals every 0.05s, and ID `0x45D` with data field containing signals encoding the angle of the gas and brake pedals every 0.01s.

The CAN standard also defines a robust error handling mechanism that is designed to prevent erroneous messages from being propagated or faulty nodes from disrupting communications. For example, if two nodes attempt to concurrently transmit different messages with the same ID, both nodes will transmit their frame until they send opposing bits simultaneously, at which point one will incur an error. If a node's error count gets too high, it will enter a "bus off" mode, meaning it cannot read or transmit messages on the bus until it is reset. See previous works [8, 66] for more details on CAN error handling.

## 2.2 CAN attacks

While lightweight, CAN lacks encryption and authentication, making it vulnerable to exploitation. There have been a number of successful attacks on vehicular CANs published in the past several years, some remote, and some requiring physical access. Koscher et al. [11] provide a comprehensive overview of CAN-based ECU vulnerabilities. Their exploration involves applications that facilitate CAN communication, such as the Unified Diagnostic Service (UDS), a standardized set of commands which can change the state of a targeted ECU or directly read and write to memory addresses. Instead of focusing on these types of applications, our attack data focuses on inherent vulnerabilities found within the CAN protocol.

The attacks surveyed here begin with the assumption of a compromised node on the bus. Cho and Shin [67] provide a well-defined adversary and attack model with terminology that is widely used. A *weakly compromised ECU* is a node that an adversary is able to silence, suspending any message transmission. A *fully compromised ECU* is a node over which the adversary has complete control, with the ability to send fabricated messages and access the node's memory. Note that the method of connecting to a vehicle's CAN via the OBD-II port is considered a fully compromised ECU in this model. Using this terminology, Cho and Shin introduce the following three general categories of attacks, which are in turn useful for describing attack sophistication and the types of IDSs that would be able to detect them.

**2.1.1 Fabrication attacks.** A *fabrication attack* [67] uses a strongly compromised ECU to inject messages with malicious IDs and Data Fields. The majority of the attacks in the CAN IDS literature fall into this category, including the following:

**DoS Attack**—Messages with ID `0x000` and an arbitrary payload are injected at a high frequency. Since ID `0x000` always wins arbitration and is not usually issued by legitimate ECUs, flooding the bus with these high priority messages prohibits legitimate messages from being transmitted. This results in a host of unusual effects, such as flashing dash indicators, intermittent accelerator/steering control, and even full vehicle shutdown. We note that this shut down is not a "limp home mode" that can occur from some attacks. Instead it

involves the vehicle ceasing functionality and rolling to a stop. The driver must remove, then reinsert the key to restart the vehicle.

**Fuzzing Attack**—(Note that the fuzzing attack here is distinct from the technique of fuzzing for vulnerability discovery. While indeed random messages are sent, it is not with the intent to reveal vulnerabilities. Nevertheless, we follow attack terminology of Cho and Shin [8] for consistency.) Messages with random IDs and arbitrary payloads are injected at a high frequency. The bus becomes occupied with mostly injected messages, displacing real messages, and resulting in similar behavior to the DoS attack. Unlike the DoS attack, injected messages may have an ID that appears in normal traffic, so receiver nodes expecting these ID messages will read and use the information in the malicious payload, causing a wide variety of unexpected results. To illustrate these effects we reference a video of a fuzzing attack [68]. There are two slight variations of this attack: some researchers inject only IDs that appear during normal traffic (e.g., [1]), while others inject arbitrary random IDs (e.g., [3]).

**Targeted ID Attack**—Messages are injected with a specific target ID and manipulated data field. When only the bits in a specific signal—that is, a select part of the 64-bit data field—are modified, we refer to this as targeting a signal, rather than an ID.

Fabrication attacks are characterized by the inherent problem of *message confliction*, described by Miller and Valasek [10],

> The biggest problem with CAN message injection is that, while attackers can inject arbitrary messages onto the bus, the original sender of the message (i.e., the legitimate ECU) is still sending legitimate messages. . .The result of the ECU continuously sending messages along side our attack messages is message confliction. From the perspective of the receiving ECU, inconsistent messages are received (and it must) decide what to do with this conflicting information.

In general, ECUs will regard the last seen data frame on a given ID; thus, to overwrite (in effect) legitimate messages with the target ID, the injected frames must occur on the bus very soon after the true frame. Not all data frames trigger an effect; simply reverse engineering a signal to inform targeted injections will often not result in the desired or any response from the vehicle. Miller and Valasek [10] provide potential techniques for side-stepping message confliction, but the desired effect for all techniques is the same—de-conflict ambient and fabricated data frames by suspending the ambient messages.

The first two fabrication attacks described (DoS and fuzzing) require almost no understanding of or reconnaissance on the target vehicle, nor do they allow for finesse in execution. On the other hand, the targeted ID attack can be more sophisticated. Targeting manipulation of specific functionality requires knowledge of at least one of the IDs' signals and requires the data field designed to have a particular effect based on the given ID's signal definitions.

Furthermore, targeted ID attacks can, similar to the first two attacks, be accomplished by *flooding* the bus, simply meaning that messages are sent at a very high frequency, although this is blatant and easy to detect. Research hackers Miller and Valesek used this tactic to successfully attack a Toyota Prius, injecting fabricated collision prevention system messages at a high frequency, causing the ABS to engage the brakes [7].

The most stealthy targeted ID attack is a *flam* attack—immediately after each target ID's legitimate message, an injected message is sent (with the same ID but manipulated data), so that the true message state is not physically realized before the spoofed message alters the car to the target state. Injected frames and true target ID frames are in one-to-one correspondence.

This type of attack was pioneered by Hoppe et al. [13], who essentially disabled a car's warning lights by sending a "lights off" frame immediately after any legitimate frame's "lights on" message was sent, resulting in the lights appearing continually off. Provided the attacker can reverse engineer the target ID's payload, only the bits involved in the targeted signal need to be manipulated.

**2.2.2 Suspension attacks.** An adversary mounting a *suspension attack* needs a weakly compromised ECU, preventing it from transmitting some or all messages [67]. For example, an adversary could suspend all messages on a particular safety-critical ID, thus disrupting other systems that rely on this constantly updated data. An example from the literature is provided by Cho and Shin [8]'s bus off attack, where error counting is manipulated until an ECU is disallowed from speaking. After this bus off attack, the ECU will not transmit its messages; hence, those AIDs would be suspended.

**2.2.3 Masquerade attacks.** Finally, the most sophisticated category, *masquerade attacks*, involve an adversary first suspending messages of a specific ID from a weakly compromised target ECU, and then using a strongly compromised ECU to inject spoofed messages with this ID at a realistic frequency, thus masquerading as the target ECU [67]. Using this more advanced strategy, a targeted ID attack can be carried out without message confliction, thus allowing for a more stealthy attack. Miller and Valasek's infamous remote Jeep Cherokee hack [10] employed a masquerade attack. Unlike the Prius [7], the Cherokee ABS system dealt with message confliction by simply turning off the collision prevention system, and thus they were unable to mount a similar fabrication attack; instead, they had to first suspend legitimate messages (in addition to a few other steps) in order to mount an attack.

In another example, Cho and Shin cleverly use a strongly compromised ECU in order to weakly compromise a target ECU by causing it to go into bus off mode, at which point they run a masquerade attack [8]. Notably, a very recent paper of Bloom [30] provides stealthier techniques for exhibiting this attack. Interestingly, if an attacker is not careful when mounting a fabrication attack, this same mechanism can result in the attacker's own strongly compromised ECU getting bussed off. In fact, we have done this on many occasions, in essence running a suspension attack on ourselves!

This previous research has shown that masquerade attacks are indeed possible, but they require ample CAN hacking expertise. Further, white-hat CAN hackers and CAN intrusion detection research communities are working independently with seemingly different skill sets toward a common goal. Thus, no real CAN data in which a masquerade attack is exhibited has been made publicly available, and the evidence from the CAN IDS research community is that most defensive researchers likely do not have the skills or resources to create such advanced attacks.

**2.2.4 Timing Transparent vs. Timing Opaque.** Unsurprisingly, these attack categories map to intrusion detection techniques that have matured alongside offensive developments. Fabrication and suspension attacks are detectable by frequency-based IDSs, which regard the timing of each ID and/or the sequential nature of IDs. Masquerade attacks require more sophisticated methods: for example, attempting to identify the sending ECU [15, 67], luring an added node to reveal itself [1], or inspecting the data field [5, 16, 64].

We define a *Timing Transparent (T.T.)* attack to be any that is hypothetically detectable using a frequency-based method: a fabrication attack, detectable by unusually fast message timing or the appearance of new IDs, or a suspension attack, detectable from unusually slow or disappearance of usually present IDs.

An attack that is *Timing Opaque (T.O.)*, on the other hand, is defined as an attack that does not disrupt normal timing or ID distributions, and thus would not be detected with a frequency-based IDS. Instead, a more sophisticated method, such as a payload-based detector

that uses the data field, or perhaps a side-channel method monitoring the physical layer, would be needed. In short, developing a comprehensive and robust IDS requires hardening (and therefore testing) against timing opaque attacks. A masquerade attack is the primary example, but other attacks that may alter the overall state of the vehicle (e.g., the "accelerator attack" in the ORNL dataset, Sec. 4.1.3, may also be included in this category.

## 3 Previous datasets

We itemize the publicly available CAN datasets with attacks, including descriptions of the attacks present, whether they are real or simulated, and the dataset benefits and drawbacks. We examined each dataset and attempted to verify the accuracy of documentation (refer to Table 1).

### 3.1 HCRL CAN Intrusion Dataset (OTIDS)

Lee et al. at the Hacking and Countermeasures Research Lab (HCRL) published this dataset as a supplement to their CAN IDS paper [1]. The dataset presented OTIDS: Offset Ratio and Time Interval based IDS, a novel IDS based on the timing of remote frame responses. The general method was to transmit remote frame requests for a given ID, time how long it took an ECU to respond to the request, and test whether this delay was anomalous—the idea being that a compromised ECU, being controlled by an adversary, would respond with an unusual delay. The published dataset from a Kia Soul contains artifacts of this method, specifically, the remote frames (which are labeled as such and thus easy to remove in preprocessing), and legitimate as well as spoofed responses (less easy to identify and remove). We note that there appear to be discrepancies in the documentation of the dataset on the website and in the paper, the figures describing the dataset, and what we find in the examination of the dataset.

**Data**—Real CAN data from one vehicle.

**Attacks**—Real. Includes DoS and fuzzing fabrication attacks. According to the authors, the dataset also includes a masquerade attack, which they call an "impersonation attack," but it does not seem as though the legitimate node (which responds to the remote frame request with the last sent message) was actually fully suspended. The first 250s of the OTIDS impersonation attack data capture are documented as ambient (no attack), and for this period we see remote frames for AID 0x164 followed shortly after by a frame with the same AID and the same data as the last seen non-remote frame with that AID (as expected). E.g.,

```
ID: 0164 000 DLC: 8 00 08 00 00 00 00 07 0f (ambient frame)

...

ID: 0164 100 DLC: 8 01 02 03 04 05 06 07 08 (remote frame)

...

ID: 0164 000 DLC: 8 00 08 00 00 00 00 07 0f (response frame)
```

Yet, during the interval of time for which the legitimate responses should have been deleted from the CAN data (>250s from start of capture), we often see the remote frame request followed by a malicious node's response (indicated by data field that does not match the last ambient frame), and a legitimate response (with the data field that does match the last ambient frame). E.g.,

```
ID: 0164 000 DLC: 8 00 08 00 00 00 00 07 0f (ambient frame)

...

ID: 0164 100 DLC: 8 01 02 03 04 05 06 07 08 (remote frame)

...

ID: 0164 000 DLC: 8 00 08 00 00 00 00 07 0f (response frame)

...

ID: 0164 000 DLC: 8 01 02 03 04 05 06 07 08 (malicious response frame).
```
Hence, this appears to be a fabrication attack in the data given, i.e., fabricating the response to a remote frame in addition to the legitimate response. This affects the AID sequence as well as the timing distribution making the attack easier to attack. These issues not withstanding, while this may be a useful simulation for testing whether their remote-frame–based IDS would detect a masquerade attack, it would not be useful for other IDSs that do not leverage this aspect of the protocol. Rough estimates of injection intervals are given.

**Benefits**— The fuzzing attack provided in this dataset is the slightly stealthier version that involves only spoofing IDs that appear in normal traffic. This is the only example of this kind of fuzzing attack in an open dataset. This is also the only open dataset with remote frames and responses.

**Drawbacks**— First and foremost, the injected messages are not labeled, and the documentation on the injection intervals is unclear and possibly incorrect. Authors indicate that in the DoS attack all `0x00` messages are injections (these take place during the entire capture), and the fuzzing and impersonation attacks start after $\sim 250$s. However, our analysis indicates that these attacks take place during the entire capture. Furthermore, this disagrees with their paper, which depicts an injected message during the fuzzing attack at 0.1565s, well before 250s. Second, as explained above, the "impersonation attack," while characterized as masquerade attack in their paper, does not seem to be a true masquerade attack since the message transmission by the legitimate node is not suspended. While it is possible that we misunderstood their documentation, our confusion on the matter and the various discrepancies we found are a testament to poor documentation. Finally, the presence of remote frame requests and responses results in small timing changes that are not usually in ambient traffic, and may be problematic for testing and training a timing-based detector. Overall, unless it used for leveraging remote frames for an IDS, this dataset is not recommended.

## 3.2 HCRL survival analysis dataset for Auto. IDS

Alongside their timing-based CAN IDS paper, Han et al. [2] at HCRL published their dataset composed of three different vehicles (Hyundai Sonata, Kia Soul, Chevrolet Spark). On each car, they collected ambient data and ran three different types of injection attacks that caused the vehicles to malfunction. Note that a different version of this dataset is also published for an IDS challenge (http://ocslab.hksecurity.net/Datasets/datachallenge2019/car), which we do not include in our survey. The original use of the dataset was to test a CAN anomaly detection algorithm using a survival analysis model.

**Data**—Real CAN data from 3 different vehicles.

**Attacks**—Real. Includes all three flooding fabrication attack types: DoS (they call this "flooding"), fuzzing, and targeted ID (flooding delivery). Each attack capture is 25–100 seconds long, and contains up to 4 five-second injection intervals. Each injected message is labeled.

**Benefits**— This is the only dataset that contains real attacks on multiple vehicles; furthermore, the same set of attacks are repeated multiple times on each one. One of the values of training and testing a timing-based IDS on multiple vehicles is illustrated in Fig 3, depicting a fuzzing attack mounted on four different vehicles (top three plots from this dataset). This illustrates how the bus load (% of time the bus is occupied with a message) can differ dramatically across vehicles, demonstrating that a timing-based IDS must be adaptable to such differences. Additionally, authors confirm that these attacks have a real effect on the vehicle. This dataset would be a good choice for training and testing a vehicle-agnostic, simple timing-based detector.

**Drawbacks**— All attacks are blatant, i.e., particularly un-stealthy, and can be detected with a very simple timing-based detector. See the top three plots of Fig 3, where the frequency of the entire bus is significantly disrupted by the exceptionally high injection frequency, which is not the case the ORNL fuzzing attack (bottom plot of Fig 3). The targeted ID attack, which they call the "malfunction" attack, is done somewhat blindly: there is no indication of what the function of the target IDs are, and the injected payloads (data fields) are chosen by either cycling through random values (on the Soul) or a single random value (on the Spark and Sonata). As for the ambient captures, only 60–90s of data are provided per vehicle, which is likely not sufficient for robust training, let alone for testing false positive rates.

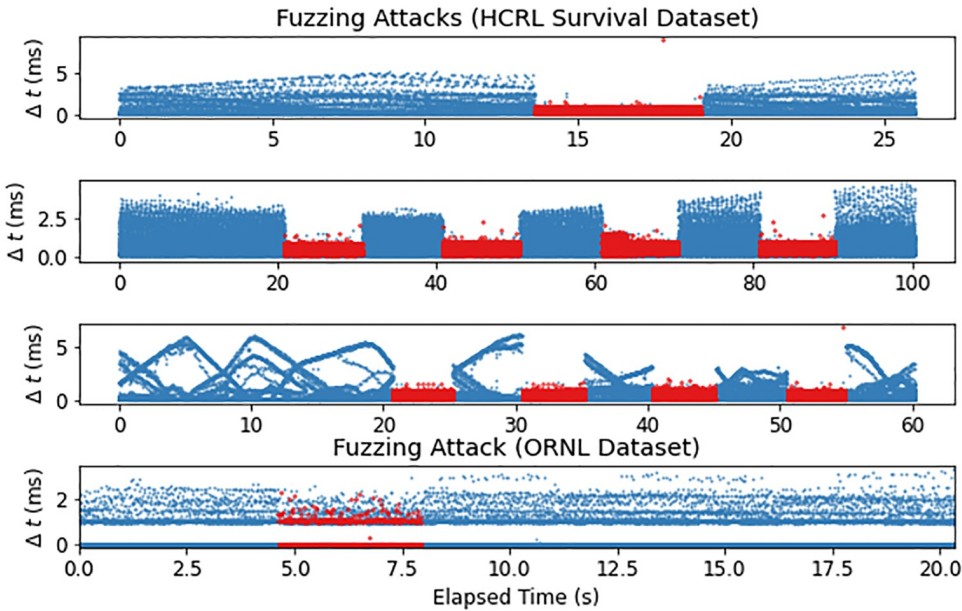

**Fig 3. The time gap between subsequent messages (all messages on the CAN capture of any ID are included) are plotted over time during fuzzing attacks on four different vehicles, with top three plots from each vehicle in the HCRL Survival Analysis Dataset, and the bottom plot from the ORNL dataset.** While the injections (in red) result in a significant disruption in the overall message timings in the HCRL dataset, the fuzzing attack in the ORNL dataset does not, and would therefore be slightly more difficult to detect using a timing-based IDS. This also illustrates that the bus load and overall message frequency distribution varies widely across vehicles.

Finally, the ambient data and attack data are in differently formatted CSVs, which is undesirable.

### 3.3 HCRL Car Hacking for intrusion detection

The Car Hacking dataset is the most recent dataset released by HCRL and was originally used by these researchers to test GIDS [3], a generative adversarial network on the AID sequence of the messages. Subsequently it was used test another deep learning IDSs [4]. The dataset includes ∼ 500s of ambient driving data, and four different attacks from a Hyundai YF Sonata.

**Data**—Real CAN data from one vehicle.

**Attacks**—Real. Includes all three flooding fabrication attack types—DoS, fuzzing, and two targeted ID attacks (flooding delivery) in which they spoof the drive gear and the RPM gauge. Each capture is upwards of 40 minutes and contains 300 attack intervals lasting 3–5 seconds. Each injected message is labeled.

**Benefits**—The attack captures are very long and contain a large number of instances per attack. Unlike the other two HCRL datasets, the function of the IDs in the targeted ID attacks is provided (drive gear and RPM gauge). This dataset seems to be the most widely used dataset in the CAN IDS literature, showing that it fills an important niche. (Unfortunately, many recent publications seem clearly to be using this dataset without citation.)

**Drawbacks**—All the attack captures contain a significant artifact of data collection that may pose a problem for researchers using this data, particularly since it is not noted in the documentation. At the conclusion of each attack (soon after the 300[th] injection), there is a large gap in messages where it appears no messages are being transmitted. This is depicted for the DoS attack in Fig 4 where there is a gap of 22s. In the other three captures containing attacks, this gap is much longer, in the order of 3000s! Berger et al. [69] also noted these jumps in timestamps. Having dealt with this issue in our own data collection efforts, we hypothesize that this message transmission gap arises from the bus going into a "stand-by" mode due to vehicle inactivity once the researchers stop injecting messages. We suggest that researchers using this dataset, particularly for a timing-based IDS, should trim the attack captures to right before the gap—at the 2328s, 2466s, 1949s, 1952s mark for the DoS, Fuzzing, Gear, RPM datasets, respectively.

While this relatively simple fix renders the dataset usable for testing, the hypothesized

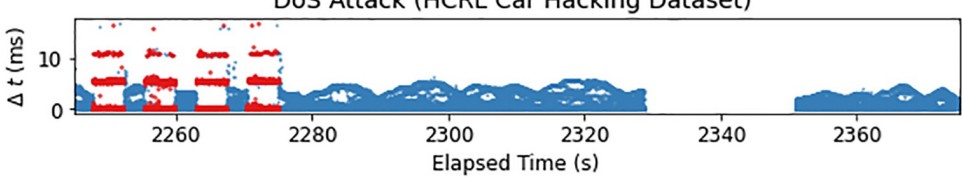

**Fig 4. HCRL Car Hacking dataset contains unintentional artifacts of data collection; in particular, in each of the four attack datasets, right after conclusion of the attack, there is a prolonged period during which no messages appear on the bus.** This depicts the end of the DoS dataset, starting from the last four injection intervals (red), followed by ∼ 53s of ambient traffic (blue), and a ∼ 22s transmission gap before ambient message resume again. Note that the first point after messages resume (with a $\Delta t \approx 22.4s$) has been omitted for scale. We hypothesize that this gap is due the CAN bus going into a "stand-by" mode due to inactivity, that is, the vehicle is not being operated and no messages are being injected.

source of the issue, namely that the car was not being driven during the attacks (which we verified by decoding the proprietary signals), poses a problem that cannot be solved in post-processing. As the car is being driven in the ambient data, the test data fundamentally differs from the training data outside of the injections, making it an unsuitable test set. Given these issues, this dataset does not seem like a good choice, even for testing a simple detector. Similar to the HCRL survival dataset, these attacks are particularly unstealthy with respect to disrupting overall bus timing (see Fig 4). Finally, ambient and attack data are in different formats (fixed width format and CSV).

## 3.4 Bosch SynCAN

Hanselmann et al. [5] at Bosch GmbH (company that created CAN), constructed a signal-translated, synthetic CAN dataset used for training and testing their CAN IDS, "CANet", an LSTM-based anomaly detector. Noting the lack of a standard, sufficient CAN benchmark dataset, Hanselmann et al. published their data for the research community. Unlike all others, this dataset contains timestamped signal values rather than the raw binary data fields; thus, it is the first and only (other than our new dataset, ROAD) dataset for evaluating a signal-based model.

**Data**– Synthetic signals (no "raw" binaries, translated time series).

**Attacks**—Simulated. Includes the following attack types: fabrication targeted ID (flooding delivery), suspension, and masquerade. Includes one capture of each attack, each containing a hundred 2–4s attack intervals. Each simulated intrusion signal is labeled.

**Benefits**—This is the only signal-based dataset (other than ROAD) which particularly important contribution since the encodings of signals into most vehicles' CAN data are unknown. It is clear from the treatment in the paper that Hanselmann et al. have true expertise in CAN protocol and data. This dataset contains the most nuanced masquerade attacks (e.g., signal values slowly drifting from a real to a target value, or replayed from normal traffic) currently available (including ROAD). Further, this is the only dataset (other than ROAD) that contains attacks targeting a single signal, rather than the full 64-bit data field. This allows for testing a very advanced, signal-based IDS.

**Drawbacks**—Synthetic data is clearly an imperfect proxy for real data; Hanselmann et al. train and test their IDS on the synthetic and real data, and remark that the former is "somewhat 'cleaner' than in the real case." As a separate issue, simulated attacks are inherently problematic since their effect on a vehicle cannot be verified. Additionally, the authors claim that all ambient data should be used for training but do not provide any additional ambient data for testing; thus, it would would difficult to test an IDS's false positive rate—an attribute of the utmost importance. Finally, since the authors do not provide a raw untranslated version of the data, many CAN detectors, which require CAN data in the usual format (time-stamped IDs with 64-bit data fields) could not use this data.

## 3.5 TU/E Auto. CAN bus intrusion dataset

This dataset was published by researchers in the Department of Mathematics and Computer Science at Eindhoven University of Technology (TU/E), who collected data from two different vehicles, an Opel Astra and a Renault Clio, and a CAN testbed, which consists of a VW instrument cluster, two Arduino boards (programmed to be a legitimate and a strongly

compromised ECU, respectively) and a joystick programmed to replicate the throttle that sends messages used by the speedometer in the instrument cluster [6]. No original use information is provided.

**Data**—Real data from two cars and synthetic data from a testbed.

**Attacks**—Simulated except for one targeted ID fabrication attack on the CAN testbed. They simulate a set of attacks on each CAN, and for all but one attack, simply augmented the recorded data in post-processing by doing the following: for fabrication attacks, they "added packets manually and adjusted timestamps accordingly"; for suspension attacks, they deleted particular frames; and for masquerade attacks, they replaced the data field of particular frames. The dataset also includes one real attack on their CAN testbed, a targeted ID fabrication attack. The joystick is used to send messages during normal traffic, and during the attack, messages with this ID are injected by the compromised ECU.

**Benefits**—This dataset includes the only diagnostic protocol attack publicly available, and the only suspension attack (simulated) in real CAN data. (Recall that SynCAN contains suspension attacks but is signal data from a simulated CAN.) The same set of attacks is available for testing on multiple vehicles/CANs.

**Drawbacks**—First and foremost, adjusting timestamps in post-processing alters data and diminishes fidelity in a critical way: message timing on the bus is dependent on each ID's frequency and priority through the arbitration process. Thus, changing message timings risks creation of a synthetic dataset that is not realistic. For the DoS attack, they simply overwrite 10s worth of frames, which is much less noisy than how real DoS attacks appear. With respect to the testbed, it is unclear how they generated ambient traffic (e.g., were they recorded and replayed from another car?), and such a testbed is an imperfect proxy for a real vehicle. Finally, attack labels are in an unstructured text file, so there is no way of programmatically reading what/when packets were injected.

### 3.6 CrySyS Lab Can-Log-Infector and ambient data

The Laboratory of Cryptography and System Security (CrySyS Lab) at Budapest University of Technology and Economics published an ambient CAN dataset (along with GPS data) from a comprehensive set of driving scenarios. They pair this data with their open-source CAN Log Infector tool (https://github.com/CrySyS/can-log-infector), which is used to manipulate data contents of the CAN log traces to simulate an attack. Using this tool, a variety of different masquerade attacks can be created in ambient CAN data by modifying only the data field of a specified target ID. Although the CAN-Log-Infector is not explicitly cited, the website hosting the tool includes a poster (https://www.crysys.hu/publications/files/KoltaiG2023CANAnomalyDetectionWithTCN-poster.pdf) presenting a temporal convolution network anomaly detector for CANs.

**Data**—Real CAN data paired with GPS data.

**Attacks**—Simulated. Includes the capability to create seven kinds of masquerade attacks. Specifically, given a start point, a target ID, and a set of contiguous bytes in the data field to target, the original value of each byte can either be replaced (by a specified constant value, random value, or increasing/decreasing sequence over each frame) or incremented (by a specified constant value or increasing/decreasing sequence over each frame). The modified copy, containing the simulated injections, does not have attacks labeled, but as all modification parameters are passed by the user, these could presumably be determined.

**Benefits**—While these authors do not provide any CAN data with attacks, the authors provide their framework for simulating a wide variety of masquerade attacks; this facilitates the creation of unlimited masquerade attacks—for example, combining different payload manipulation techniques simultaneously on different IDs, in any CAN data. As this software is open source, it could be extended to add new attacks. This is the only dataset with real CAN data that allows for injections that vary continuously over time and is the only dataset (other than our new dataset, ROAD) that allows for modifying only part of a data field; thus, it enables attacks targeting signals (select bits of a payload). Other than ROAD, this is the only dataset furnished with descriptions of the driver's actions during ambient captures, which is highly valuable when for training and testing an IDS.

**Drawbacks**—As attacks are added in post-processing, there is no guarantee that these attacks would actually affect vehicle function. Moreover, these attacks are completely blind to the function and signal mapping of particular target IDs. There are also logical problems with Can-Log-Infector's implementation, most notably that the data field can only be modified at the byte level (i.e. replacing characters in hex representation) and all selected bytes must change uniformly. Thus, signals that do not exactly fill a set of bytes cannot be solely targeted (recall that payloads are composed of several signals of varying lengths and positions whose bits often cross byte boundaries). Furthermore, incrementing whole bytes means multi-byte signals will vary in a highly discontinuous manner. Finally, the method for specifying the injection interval is rather irksome—rather than specifying a starting timestamp, the user passes "a value between 0 and 1 (indicating) the ratio when the attack should start regarding the full length of the capture," and the attack end point cannot be specified.

## 4 Introducing the ROAD dataset

The ROAD dataset https://zenodo.org/records/10462796 DOI: 10.5281/zenodo.10462795) consists of 33 attack captures totalling about 30m, and 12 ambient captures totalling about 3h. See Table 5; attacks are described in more detail in Sec. 4.1 syntactic metadata appears in the Appendix 6.2 and further descriptions are provided in the documentation published with our dataset. Using the CAN-D algorithm [61] to translate the raw CAN data into signals, we also provide a signal-translated version for 17 of the 33 attack captures and the all ambient captures. See description in Appendix 6.3.

We collected CAN data using SocketCAN software on a Linux computer with a Kvaser Leaf Light V2 connecting to the OBD-II port. All data is from a single vehicle, the make/model of which our organization will not allow us to disclose, and with year of manufacture in the mid 2010s. The published data has been obfuscated in a way that maintains the anonymity of the vehicle, while preserving nearly all important aspects of the data for IDS research—details of obfuscation reside in 4.2. For all attacks, the vehicle was actively being driven on a dynamometer. Notably, each attack's intended alteration of vehicle functionality was physically verified and documented—an advantage of using real CAN data and a dynamometer. Ambient data was collected both on a dynamometer and on roads, while performing a variety of normal and sometimes unusual driving activities (e.g., opened door while driving). This allows training/ testing with anomalies to both increase realism and allow investigation of false positives. Metadata with details on the activities in each capture and on the physical effect of each attack are provided. Examples appears in Fig 6.

The breadth of ambient and attack data in the ROAD dataset is designed for testing a variety of detectors—using different features (e.g., timing, payload bits, or signals), and modeling different characteristics (e.g., frequency, entropy, continuity, or correlation). Fig 5 illustrates

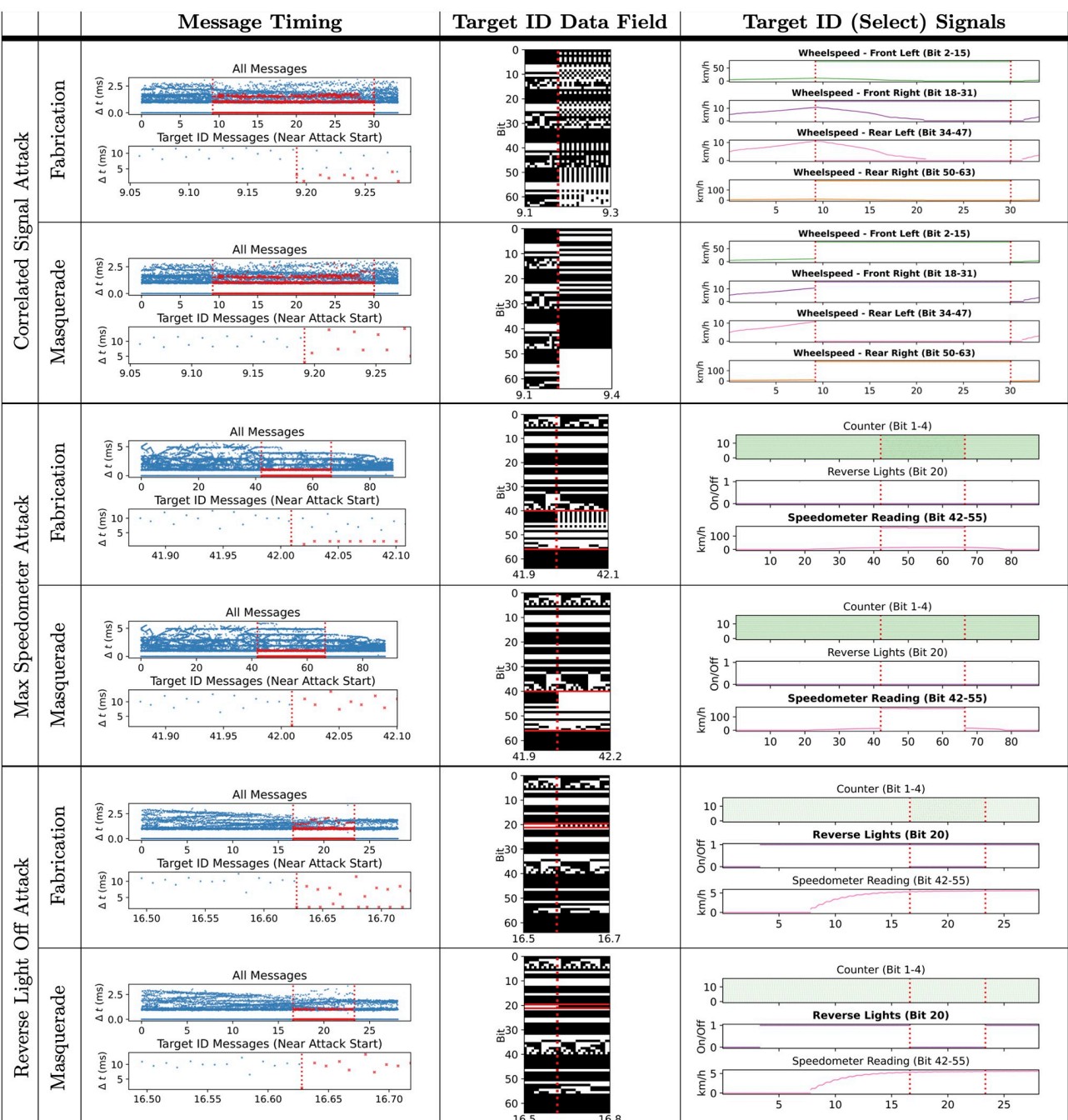

**Fig 5. Depiction of six of the targeted ID and masquerade attacks in the ORNL dataset.**

characteristics of the data, with visualizations of three different features (timing, payload, signals) of six attack captures in our dataset. While ROAD's fuzzing, fabrication, and suspension attacks provide T.T. attacks of increasing stealth, the Accelerator Attack—a result of a newly discovered and disclosed vulnerability—and many simulated masquerade attacks are T.O. For developing detector using payloads, the Correlated Signal, Max Speedometer, and Reverse Light attacks all entail discontinuities or break correlation in CAN signals.

### 4.1 ROAD dataset attacks

Attack captures detailed in order of expected detection difficulty.

**4.1.1 Fuzzing attack.** We mounted the less stealthy version of the fuzzing attack, injecting frames with random IDs (cycling IDs in order from `0x000` to `0xFF`) with the maximum payload `0xFFFFFFFFFFFFFFFF`) every.005s, as opposed to the more stealthy version which only injects IDs seen in ambient data. This attack is designed to be easy to detect. There were many physical effects of this attack, for example: the accelerator pedal became ineffective; the dash lights and headlights were illuminated; and the seat positions moved (see Section 2.2).

**4.1.2 Targeted ID fabrication and masquerade attacks.** We performed targeted ID fabrication attacks using the flam delivery, meaning a message is injected immediately after a legitimate message with the target ID is seen. As discussed in Section 2.2, the flam technique allows for dynamic injection; that is, the legitimate ID message is read, only the bits corresponding to the target signal are modified with malicious values, and then this spoofed message is injected. When only part of the message is modified, we refer to this as targeting a signal, rather than an ID. Designing these attacks required reverse engineering of signals for this vehicle, which we completed using CAN-D [61] signal reverse engineering algorithm and manually verifying the results. The targeted ID fabrication attacks and masquerade attacks are as follows:

- **Correlated Signal**—The single ID communicating the four wheels' speeds (each is a two-byte signal) is injected with four false wheel speed values that are all very different. The effect, supposing the injected frames are at least as fast as the ambient frames with that ID (the case in this dataset), is that the accelerator has no effect on the vehicle. This loss of control begins immediately and throughout the injection period. In some instances the car required restart to return to normal functionality.

- **Max Speedometer**—The one-byte speedometer signal is targeted by sending (`0xFF`), causing the speedometer to falsely display a maximum value.

- **Max Engine Coolant Temperature**—We target the engine coolant signal (one byte), modifying the signal value to be the maximum (`0xFF`). The physical effect is an "engine coolant too high" warning light on the dash illuminates.

- **Reverse Light**—A one-bit signal communicating the state of the reverse lights (on/off) is targeted. We perform two slight variations of the attack, manipulating the value to off (on), while the car is in Reverse (Drive), respectively. The effect is that the reverse lights do not reflect the gear (Drive/Reverse).

For all of these targeted ID attacks, we provide two versions of the same CAN data captures: the original fabrication attack, and a version slightly modified in post-processing to simulate a masquerade attack. Refer to the Table 5, which itemizes the fabrication/masquerade pairs. A subset of the fabrication/masquerade pairs attacks are visualized in Fig 5.

The fabrication attack versions are the original altered capture, including both the legitimate target ID frames and the injected frames. Because these are real, physically verified attacks with the minimally occurring injected frames (due to the flam delivery), they provide perhaps the best (i.e., most stealthy/most difficult to detect), current, public data for testing frequency-based IDSs. That is, most fabrication attacks in public datasets involve many injected frames between the ambient vehicle frames with the same ID, while the flam delivery used for ROAD's targeted ID attacks have a single injected frame between ambient frames of the same ID. As at least one injected message occurring after each legitimate message is needed to manifest the desired physical effect; hence, the flam delivery, with only one such message, is the most stealthy possible.

Using the fabrication captures we produce simulated masquerade attacks by removing the legitimate target ID frames preceding each injected frame to provide more advanced versions. In effect, this removes message confliction in the data, making it appear as though only the spoofed messages are present during the injection interval. With this masquerade dataset, frequency-based approaches will almost certainly fail to provide accurate detection. It is important to note that while the masquerade aspect is simulated through post-processing, this means of alteration avoids problematic issues with synthetic data. Namely, the effect of the attack on the vehicle was physically verified; every message appearing in the data was actually seen by the car in the order it appears in the data; and no aspect of CAN protocol was violated. As discussed in the introduction, there are no publicly available, real CAN data captures with real masquerade attacks, and the hacking skill required to implement such an attack on a real vehicle seems to be preventing CAN IDS researchers from implementing such an attack. This provides the highest fidelity alternative possible.

Considering the top two rows of Fig 5 we can clearly see the difference between the fabrication attack and corresponding simulated masquerade attack. In the top row (fabrication attack), all four wheelspeed signals appear to move continuously but are joined by a second curve—the injected value of that signal—during the attack interval; whereas, in the second row (masquerade attack) continuity of the signal values is broken and only the injected values appear during the attack interval.

**4.1.3 Accelerator attacks.** We have responsibly disclosed this vulnerability to the OEM, and will not disclose details of how to implement this attack. We do not include the CAN data during the exploit. After the exploit, the effect is that the vehicle is in a state that has less control by the driver as follows: when put into Drive gear, the vehicle accelerates to a fixed speed and then holds this speed (regardless of accelerator pedal position or cruise control setting); when in reverse, the vehicle accelerates to a (different) fixed speed and holds this speed (regardless of accelerator pedal position or cruise control setting); cruise control is disabled; touching the brake pedal results in the acceleration ceasing and the brakes engaging normally; when the brake is released, the vehicle commences accelerating as described above. The Accelerator Attack captures have no injected messages, but simply record the CAN data when the vehicle is in this state. Discrepancies exist between the vehicle's actions and the driver's inputs, e.g., acceleration occurs regardless of the accelerator pedal position.

## 4.2 Obfuscation

While other public CAN datasets provide information on the make, model, and year of the vehicles attacked, it would be irresponsible, given our previous disclosure, to release such information. Furthermore, we have taken steps to obfuscate the CAN data in such a way as to preserve the characteristics necessary for CAN IDS development, while ideally preventing users from knowing the make, model, and year of the vehicle. Below we itemize the augmentations performed on the data to preserve anonymity:

- Absolute timestamps are shifted uniformly by a scalar.

- Arbitration IDs that were constant, aperiodic, or periodic with frequency under 0.1 Hz (less than one frame per ten seconds) were replaced with the "filler message" `FFF#0000000000000000` (ID#Data in hex) and same relative timestamp.

- Messages on reserved IDs (greater than `0x700`: e.g., diagnostic messages) have been removed.

- IDs have been anonymized in such a way that *arbitration order/priority is not preserved*. There is a one-to-one mapping between the original and the anonymized IDs for a given vehicle (not including the "filler messages" under ID `0xFFF`). For example, if ID `0x10` is converted to ID `0x821` in an anonymized log, the same is true for all logs.

- Data fields have been scrambled in such a way that signals have been preserved, and fields are scrambled in a consistent way for each ID; e.g., if the first byte is moved to the end of the field for ID `0x10`, it will be shifted this way in all messages from ID `0x10`.

## 4.3 ROAD drawbacks

The dataset includes only attacks on the vehicle while on the dynamometer, and data collected on a dynamometer will have subtle differences than that collected on actual road driving. (Notably, ambient dynamometer data is provided.) ROAD contains data from a single vehicle. The timestamps are accurate to 100$\mu s$, which may not offer a high enough resolution for testing certain time-based detectors. Attack intervals are labeled rather than labeling each message. Obfuscation of the data intentionally did not preserve order of the IDs; as IDs encode priority, some IDSs may require this information. This dataset does not include information useful for physical-layer-based intrusion detection. Finally, ROAD's masquerade attacks rely on a small amount of simulation. (To our knowledge no un-simulated masquerade attack data is available, and this gap is a hindrance for the community.)

## 4.4 ROAD advantages

The use a dynamometers allows driving while attacking the vehicle. Ambient dynamometer driving dataset includes a variety of activities such as reversing, accelerating, and opening/closing doors, and this dataset can be useful for training detectors with realistic activities, including non-hacking anomalies, and/or testing for false positives. ROAD provides blatant to very subtle attacks for development and testing. By using the flam technique (one injected message right after each ambient message, and only necessary bytes in the data field manipulated), ROAD dataset provides the stealthiest possible fabrication attacks. While Miller and Valasek have exhibited masquerade attacks on real vehicles (even remotely), to our knowledge, no available CAN data contains non-simulated attack. ROAD's simulated masquerade attack provides the best possible alternative to a real masquerade attack. The Accelerator Attacks are unique captures of CAN data from a vehicle with only the vehicle's ECUs transmitting messages, but in a compromised manner—there is no disruption of the timing of the CAN messages. To our knowledge, no detectors claiming ability to detect T.O. attacks have been tested on real data; hence, these captures fill a gap in the available data for testing detectors. Further, for detectors that rely on the CAN data frame (payload).

ROAD provides many examples of attacks that cause discontinuities in signals or discrepancies between normally highly correlated signals. All attacks in the ROAD dataset have been physically verified to affect the vehicle's functionality. ROAD provides metadata documenting driving activities and attack effects of each capture.

## 4.5 Case studies using ROAD

A main contribution of the ROAD dataset is to serve the CAN IDS research community to facilitate appropriate benchmarking and needed comparability in the CAN IDS research field. This is reflected in recent literature. Below is a brief overview of the current utilization of ROAD.

ROAD allows researchers to compare and contrast different techniques in the CAN IDS realm for benchmarking as seen in [23, 70]. Researchers in [70] compare the evaluation of a variety of anomaly-based CAN IDS methods against the ROAD dataset. The authors mention that the ROAD dataset stands-out compared to other datasets because it consists of observations with the stealthiest targeted ID fabrication. Sharmin et al. evaluated four statistical (ID sequences, entropy-based, Hamming distances, frequency-based) and two ML-based (OCSVM, isolation forest) CAN IDS algorithms against ROAD. An in-depth evaluation of the IDS methods is provided including training time, testing time on different attacks, precision, recall, accuracy, F1-score, balanced accuracy (bACC), informedness (BM), markedness (MK), and Matthews Correlation Coefficient (MCC). From these metrics, the investigators discovered that the entropy-based algorithm was most effective against fuzzing and targeted ID attacks. Authors provide deep comparison and analysis of the performance of each algorithm however, the overlying point is that the ROAD dataset enabled for this comprehensive evaluation. Blevins et al. benchmark four time-based IDSs (mean inter-message time, binning, fitting Gaussian distribution, and kernel density estimation) against the ROAD dataset [23]. The researchers discovered that the two distribution-agnostic achieved the highest F1 scores while the distribution-based approaches were limited in comparison, suggesting that heuristic approaches outperform methods that explicitly relay on $p$-value thresholds.

Another key aspect of introducing the research community to the ROAD dataset is that it provides a quality platform to test novel IDS architectures. In [37, 71–78], researchers use the ROAD dataset for evaluation purposes from a novel IDS method. Jin et al. combine oversampling, outlier detection, and metric learning for intrusion detection and evaluate their model on ROAD (and other datasets) [71]. Suhail showcase a gamification (attacker vs defender) framework for assessing physical cyber security of digital twins [72]. In [73], a study investigates the use of a context aware IDS for detecting cyberattacks on the CAN bus and use the ROAD dataset for training. The potential of embedding an IDS that utilizes characteristic functions is proposed in [79], where researchers evaluate the cybersecurity framework on ROAD. In [77], researchers utilize the ROAD dataset to validate a model called "Deep Evolving Stream Clustering- IDS" or DESC-IDS. Cheng et al. propose the model as a means of anomaly detection capable of reducing data complexity for constructing spatial-temporal features and exposing attacks. Shahriar et al. compare evaluation of a model called CANShield between ROAD and SynCAN datasets [37]. Researchers show anomaly scores and ROC curves for the attacks occurring with the ROAD dataset. Moriano et al. [78] propose a forensic framework for detecting masquerade attacks. Authors demonstrate the results from the study indicate high effectiveness of detecting attacks and the potential of utilizing said framework for real-time IDS. These pieces of literature demonstrate the impact that the ROAD dataset can have for enabling novel model evaluations.

ROAD also presents as a hub for researchers to reference the taxonomy of CAN data. Systematic and survey literature has recently been published citing the ROAD data [51, 80–82]. Some studies make claim that the ROAD dataset is the most comprehensive and realistic open CAN dataset available for evaluating and comparing CAN IDSs for attacks [51, 81]. Other research has referenced the quality of the dataset, used it to establish definitions within the CAN IDS research community, or cited the work as an establishment of research standards [21, 83–90].

Finally, a few studies have plans to utilize the ROAD dataset for future investigations. Agbaje et al. plan to utilize the ROAD dataset for benchmarking in the future [91]. Papadopoulos argue for the use of Named Data Networking for a solution to automotive network issues (compared to CAN local interconnect network, low-voltage differential, etc.) and plan

on implementing it on ROAD in the future [92]. The availability of the ROAD dataset is what enables these works to streamline and push the frontier of this research area forward with ease.

## 5 Conclusion

In this paper, we identify two troubling trends for the CAN IDS research community: the lack of comparability of CAN IDS methods and an inability to test IDS approaches targeting subtle, more advanced attacks. By providing the first comprehensive guide to publicly available CAN data, we contribute a single source for future researchers to consult when needing to identify the best public dataset for their developments. Further, we contribute the new ROAD dataset, containing real CAN data with a wide variety of attacks, designed to allow testing of the multitude of different techniques arising in the literature. Many advancements/gaps are made/bridged by ROAD—See ROAD's Advantages 4.4. Notably, gaps that still exist in the CAN IDS data include: real masquerade attack data; more real signal-translated CAN data with attacks; CAN data with physical-level characteristics (e.g. voltage). Finally, it is outside the scope of this paper to use the dataset to test IDS methods. Such examples are appearing in the literature, e.g., [23].

## 6 Appendix

### 6.1 Description of masquerade attacks

Fig 5 shows both the Timing Transparent (fabrication, with message confliction) and Timing Opaque (masquerade, without message confliction) versions for three attack types. The *x*-axis of all plots are elapsed time (s), and the red dashed lines demarcate the attack interval. The three main columns visualize different aspects of each of the six attacks: **Message Timing**: Inter-message arrival time (ms) between all messages shown in the Top *All Messages* subplot, and between only the target ID messages in the bottom *Target ID Messages (Near Attack Start)* subplot, which zooms in to 15s before to 20s after the attack start. Blue dots/ red x's indicate legitimate/injected messages. Compare the six *All Messages* (Top) subplots with Fig 3 to see overall bus timing is nearly undisturbed, whereas previous attacks are blatant. The six *Target ID Messages* (bottom) subplots illustrate that the fabrication attacks (using flam injection delivery) cause unusually short inter-message times for the target ID, while masquerade attacks do not cause perceptible timing changes. **Target ID Data Field**: Time series of 64-bit binary data during the time period near the attack start (black denotes 1s, white denotes 0s). If only part of the message was altered (i.e., one target signal), the section of altered bits are delimited with red solid lines. Through visual inspection, fabrication attacks are more obvious due to message confliction, and both fabrication and masquerade attacks are more noticeable when the entire message is targeted (e.g., Correlated Signal Attack), rather than just a single signal. **Target ID Signals**: The time series of signals in the target ID message are depicted, annotated with signal names and bit ranges, which are made boldface for target signals (note not all non-target signals in the message may be shown). Notice the Max Speedometer and Reverse Light Off attack target different signals in the same ID. While even the masquerade versions of the first two attack types are somewhat visually identifiable at the signal level due to discontinuities and extreme values, the Reverse Light Off Attack targeting a 1-bit signal is difficult to discern without understanding more complex signal relationships or by examining signals in other messages.

## 6.2 Syntactic description

All of the CAN data files are logged using the standard can-utils (https://github.com/linux-can/can-utils) candump format:

$$\underbrace{(1569510697.667343)}_{\text{Unix Timestamp}} \quad \underbrace{\texttt{can0}}_{\text{Channel}} \quad \underbrace{\texttt{5E1}}_{\text{ID (hex)}} \quad \underbrace{\texttt{893FE0070A000080}}_{\text{Data Field (hex)}}$$

While timestamps are reported with a precision of $1\mu s$, the hardware used to collect this data (a Kvaser Leaf Light V2) only guarantees an accuracy of $100\mu s$. Note that all data fields in these logs contain the full 8 bytes, which we padded with zeros if necessary. The channel is always can0, so this column can be dropped. We provide metadata (in JSON format) for each capture, including a general description of driving activities, the length of the capture in seconds, and whether or not the car was on the dynamometer. For attack captures, we also include whether the capture was modified (i.e., masquerade attacks), the injection ID and data field, and the interval of injection (start, end) corresponding to the time of the first/last injected message in elapsed seconds. Importantly, we do not label individual messages as attack/normal, because the software we used to collect did not have that capability. However, with injection ID, data, and intervals, these can be labeled in post-processing fairly easily. Examples of the provided metadata are shown in Fig 6.

We use a wildcard character "X" in the injection_data_str field to indicate that the byte in the given position was not modified in the injection when only one signal in the data field is targeted. Similarly, "X" in the injection_id field indicates that no particular ID was targeted, which is only the case in the fuzzing attack. For the accelerator attack, the injection_id and injection_data_str are null, and the injection interval is just the start and end time of the capture (Note that all of these details are included in the full documentation).

The translated time series are represented in CSV format, following a similar schema as SynCAN [5], the other signal translated dataset. Specifically, the CSV files have the following columns: Label, ID, Time, and Signal-<i>-of-ID. Labels are either 0 (benign) and 1 (attack), and all the entries in the ambient captures are labeled 0. Each of the signals within an ID is named based on the index they have when translated, i.e., $i \in 0, 1, \ldots, N_{ID} - 1$, where $N_{ID}$ is the maximum number of signals in a particular ID. We added a metadata file for each of the logs describing the details of the CSV files.

```
ambient_dyno_drive_basic_short:
    {
    description: "start from
        park; basic drive
        activities (e.g., drive
        ; accelerate; brake;
        reverse; ect.)"
    elapsed_sec: 444.75061,
    on_dyno: True }
```

```
correlated_signal_attack_1:{
    description: "start from
        driving; accelerate;
        start injecting; car
        rolls to stop; stop
        injecting; accelerate"
    elapsed_sec: 33.101852,
    injection_id: "0x6e0",
    injection_data_str:
        "595945450000FFFF",
    injection_interval:
        [9.191851, 30.050109],
    modified: False,
    on_dyno: True}
```

**Fig 6. Snippet of metadata for two example captures, with an example of ambient (left) and attack (right) entries.**

## 6.3 Time series (Signal) translation

We translated the complete set of 12 ambient captures and 17 of the 33 attack captures. Our goal in providing the time series signal translation is to provide researchers with an open and realistic dataset for benchmarking signal-based IDS methods. For that reason we excluded fabrication attack captures that are easily detected without consulting the frames' data fields (they can be detected by monitoring IDs and their timing). We provide the translated signals of the data fields in the masquerade and accelerator attack captures, as these attacks likely require analytics that regard the CAN frames' data fields.

We used CAN-D [61] to translate the signals and generate DBCs, using the heuristic signal boundary classifier (see Sec. III.A of the CAN-D paper) as it exhibited superior performance. To generate the DBC, we concatenated all of the ambient captures from when the car was on the dynamometer (10 out of 12 ambient captures), and used this as input to the CAN-D pipeline. We used the ambient captures generated in the dynamometer because all of the attacks were executed in dynamometer conditions. The generated DBC output is also made available. Note that "Step 4 Physical Interpretation" (see Sec. III.D of the CAN-D paper) of the CAN-D pipeline requires matching translated signals to time series of an extra sensor (e.g., using on-board diagnostics) to linearly scale the extracted signals to appropriate units and know their label (e.g., wheels speed in km/h). The released signal-translated ROAD data does not include the physical interpretation of the signals as no diagnostic queries or other external sensor time series are included.

For illustration purposes, Fig 5 contains the bitstreams (center column) and the translated time series (right column) of time windows containing the attack start and the full attack period respectively. Dashed red vertical lines represent the injection interval of the attack. Finally, note that we independently reverse engineered the physical interpretation of the signals for Fig 5; hence, the table depicts signal labels and units unavailable in the released data and the plots are of linearly translated values from what is released.

## Acknowledgments

Thanks to Gedare Bloom for pointing out timestamp issues in an early version of the ROAD dataset. Thanks to Suzanne Parete-Koon and Ross Miller for assistance in posting the dataset online. Thanks Stacy Prowell and John Baston for helping us polish this document.

## Author Contributions

**Conceptualization:** Miki E. Verma, Robert A. Bridges, Pablo Moriano.

**Data curation:** Miki E. Verma, Michael D. Iannacone, Samuel C. Hollifield.

**Formal analysis:** Miki E. Verma, Robert A. Bridges, Frank L. Combs.

**Funding acquisition:** Robert A. Bridges.

**Investigation:** Miki E. Verma, Robert A. Bridges, Michael D. Iannacone, Samuel C. Hollifield, Pablo Moriano, Steven C. Hespeler, Bill Kay, Frank L. Combs.

**Methodology:** Miki E. Verma, Robert A. Bridges, Samuel C. Hollifield, Pablo Moriano, Steven C. Hespeler, Frank L. Combs.

**Project administration:** Robert A. Bridges.

**Resources:** Robert A. Bridges.

**Software:** Miki E. Verma, Robert A. Bridges, Michael D. Iannacone, Samuel C. Hollifield, Bill Kay.

**Supervision:** Robert A. Bridges, Pablo Moriano.

**Validation:** Miki E. Verma, Robert A. Bridges, Michael D. Iannacone, Samuel C. Hollifield, Pablo Moriano, Bill Kay, Frank L. Combs.

**Visualization:** Miki E. Verma, Robert A. Bridges, Pablo Moriano, Steven C. Hespeler, Bill Kay.

**Writing – original draft:** Miki E. Verma, Robert A. Bridges, Michael D. Iannacone, Samuel C. Hollifield, Pablo Moriano, Steven C. Hespeler, Bill Kay, Frank L. Combs.

**Writing – review & editing:** Miki E. Verma, Robert A. Bridges, Pablo Moriano, Steven C. Hespeler.

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
