## [Decision Letter · Decision Letter 0]

18 Sep 2023

PONE-D-23-22932A comprehensive guide to CAN IDS data and introduction of the ROAD datasetPLOS ONE

Dear Dr. Moriano,

Thank you for submitting your manuscript to PLOS ONE. After careful consideration, we feel that it has merit but does not fully meet PLOS ONE’s publication criteria as it currently stands. Therefore, we invite you to submit a revised version of the manuscript that addresses the points raised during the review process. Please submit your revised manuscript by Nov 02 2023 11:59PM. If you will need more time than this to complete your revisions, please reply to this message or contact the journal office at plosone@plos.org. Please include the following items when submitting your revised manuscript:A rebuttal letter that responds to each point raised by the academic editor and reviewer(s). You should upload this letter as a separate file labeled 'Response to Reviewers'.A marked-up copy of your manuscript that highlights changes made to the original version. You should upload this as a separate file labeled 'Revised Manuscript with Track Changes'.An unmarked version of your revised paper without tracked changes. You should upload this as a separate file labeled 'Manuscript'.

We look forward to receiving your revised manuscript.

Kind regards,

Vincent Omollo Nyangaresi, Ph.D

Academic Editor

PLOS ONE

6. Please remove your figures from within your manuscript file, leaving only the individual TIFF/EPS image files, uploaded separately. These will be automatically included in the reviewers’ PDF.

Reviewers' comments:

Reviewer's Responses to Questions

**Comments to the Author**

1. Is the manuscript technically sound, and do the data support the conclusions?

Reviewer #1: Yes

Reviewer #2: Yes

2. Has the statistical analysis been performed appropriately and rigorously? 

Reviewer #1: I Don't Know

Reviewer #2: Yes

3. Have the authors made all data underlying the findings in their manuscript fully available?

Reviewer #1: Yes

Reviewer #2: Yes

4. Is the manuscript presented in an intelligible fashion and written in standard English?

Reviewer #1: Yes

Reviewer #2: Yes

5. Review Comments to the Author

Reviewer #1: 1- The introduction explains well the work environment and the proposed method, but it lacks the conclusions reached by the researcher based on specific criteria. In addition, the line alignment is not formatted

2- Many types of CAN attacks are not referenced. Such as Fabrication Attacks and Suspension Attacks. Please pay attention to adding references to the paragraphs extracted from their original references.

3- In Previous Datasets, it would have been better to shorten much of the narrative by adding tables that express the differences between the types of Datasets according to the classifications mentioned. Thus, we provide easy follow-up and comparison

4- The scientific term is mentioned with its abbreviation at the first appearance only

5- The researcher did not present a comparison between ROAD Dataset and other previous types.

6- Organizing references according to a unified format

Reviewer #2: The authors present CAN IDS dataset. Here are some comments:

1- In abstract, the criteria used to present a dataset and the types of attacks must be mentioned and the types of attacks must be mentioned

2- On page 2, Introduction section, “As a result, proposed detection techniques are often not tested on appropriate data due to lack of availability.”, What is the evidence for this claim?

3- The types of attacks and their names must be mentioned in the Introduction Section.

4- From line 55-70, lacks more modern references.

5- In line 91, “40 CAN IDS papers surveyed”, these references must be cited

6- Please add surveys for recent years.

7- The propriety CAN signal problem must be added to the article, even if the addition is in brief with tables.

8- On page 5, Section “Problem Addressed”. Does the current CAN IDS fail to prevent attacks in the real world? Examples should be mentioned.

9- On page 6, line 189-194, what is the difference from Reference 28, is that its data is considered standard.

10- In contributions, you must briefly mention the environment in which the data was trained for three hours

11- On page 9, 2.1 Can Protocol, An example must be given to illustrate the transfer of messages between ECUs, indicating the fields of payload.

12- Recent references to the aforementioned attacks must be cited, such as:

https://ieeexplore.ieee.org/abstract/document/9274321

https://www.mdpi.com/2079-9292/12/17/3688

https://www.mdpi.com/2224-2708/11/3/55

13- On page 10, the difference between DOS and Fuzzy attacks must be clarified. Are there examples in the real world of these attacks?

14- On page 12, the mechanism of the suspension attack is unclear.

15. Section 3 Previous Datasets, Usage area must be added for each attack

16. Section 4. Introducing the ROAD Dataset, At least mention the name of the vehicle manufacturer, but not its model, according to your organization’s regulations.

17. On page 20, Why was 0.005 seconds chosen?

18. A table must be added in the results section showing the difference between the presented database and its predecessors

6. PLOS authors have the option to publish the peer review history of their article (what does this mean?). If published, this will include your full peer review and any attached files.

Reviewer #1: No

Reviewer #2: **Yes: **Zaid Ameen Abduljabbar

---

## [Author Response · Author response to Decision Letter 0]

10 Nov 2023

I have attached an editor and reviewer letters independently with this submission.

---

## [Decision Letter · Decision Letter 1]

20 Nov 2023

PONE-D-23-22932R1A comprehensive guide to CAN IDS data and introduction of the ROAD datasetPLOS ONE

Dear Dr. Moriano,

Thank you for submitting your manuscript to PLOS ONE. After careful consideration, we feel that it has merit but does not fully meet PLOS ONE’s publication criteria as it currently stands. Therefore, we invite you to submit a revised version of the manuscript that addresses the points raised during the review process.

We look forward to receiving your revised manuscript.

Kind regards,

Vincent Omollo Nyangaresi, Ph.D

Academic Editor

PLOS ONE

Reviewers' comments:

Reviewer's Responses to Questions

**Comments to the Author**

1. If the authors have adequately addressed your comments raised in a previous round of review and you feel that this manuscript is now acceptable for publication, you may indicate that here to bypass the “Comments to the Author” section, enter your conflict of interest statement in the “Confidential to Editor” section, and submit your "Accept" recommendation.

Reviewer #1: All comments have been addressed

Reviewer #2: (No Response)

2. Is the manuscript technically sound, and do the data support the conclusions?

Reviewer #1: Yes

Reviewer #2: Partly

3. Has the statistical analysis been performed appropriately and rigorously? 

Reviewer #1: Yes

Reviewer #2: Yes

4. Have the authors made all data underlying the findings in their manuscript fully available?

Reviewer #1: Yes

Reviewer #2: Yes

5. Is the manuscript presented in an intelligible fashion and written in standard English?

Reviewer #1: Yes

Reviewer #2: Yes

6. Review Comments to the Author

Reviewer #1: (No Response)

Reviewer #2: The article is weak in analysis and has not yet been improved despite being directed extensively by comments. No significant contribution was highlighted. It also lacks references that shed light on a broad scope of the problem. Some references have been suggested that researchers have not addressed

7. PLOS authors have the option to publish the peer review history of their article (what does this mean?). If published, this will include your full peer review and any attached files.

Reviewer #1: No

Reviewer #2: No

---

## [Author Response · Author response to Decision Letter 1]

30 Nov 2023

We have attached our responses to the editor (cover letter) and reviewers (response to reviewers).

---

## [Decision Letter · Decision Letter 2]

21 Dec 2023

A comprehensive guide to CAN IDS data and introduction of the ROAD dataset

PONE-D-23-22932R2

Dear Dr. Moriano,

We’re pleased to inform you that your manuscript has been judged scientifically suitable for publication and will be formally accepted for publication once it meets all outstanding technical requirements.

Kind regards,

Kamran Siddique

Academic Editor

PLOS ONE

Reviewers' comments:

Reviewer #1: All comments have been addressed

Reviewer #2: All comments have been addressed

2. Is the manuscript technically sound, and do the data support the conclusions?

Reviewer #1: Yes

Reviewer #2: Yes

3. Has the statistical analysis been performed appropriately and rigorously? 

Reviewer #1: Yes

Reviewer #2: Yes

4. Have the authors made all data underlying the findings in their manuscript fully available?

Reviewer #1: Yes

Reviewer #2: Yes

5. Is the manuscript presented in an intelligible fashion and written in standard English?

Reviewer #1: Yes

Reviewer #2: Yes

6. Review Comments to the Author

Reviewer #1: (No Response)

Reviewer #2: After the second round of revisions, the authors have fully completed all required revisions, and I recommend accepting the article

---

## [Editor Report · Acceptance letter]

9 Jan 2024

PONE-D-23-22932R2 

PLOS ONE

Dear Dr. Moriano, 

I'm pleased to inform you that your manuscript has been deemed suitable for publication in PLOS ONE. Congratulations! Your manuscript is now being handed over to our production team.

Kind regards, 

on behalf of

Dr. Kamran Siddique 

Academic Editor

PLOS ONE